health and disease and epidemiology, ecology

SARS-CoV-2, COVID-19, time-varying reproduction number

**Author for correspondence:**
Jue Tao Lim
e-mail: ephljt@nus.edu.sg

[†]These authors contributed equally.

# Revealing regional disparities in the transmission potential of SARS-CoV-2 from interventions in Southeast Asia

Jue Tao Lim[1], Borame Sue Lee Dickens[1,†], Esther Li Wen Choo[1,2,†], Lawrence Zheng Xiong Chew[1,3,†], Joel Rui Han Koo[1], Clarence Tam[1], Minah Park[1] and Alex R Cook[1]

[1]Saw Swee Hock School of Public Health, National University of Singapore and National University Health System, Singapore
[2]Department of Biological Sciences, Faculty of Science, National University of Singapore, Singapore
[3]Department of Geography, Faculty of Arts and Social Sciences, National University of Singapore, Singapore

(iD) JTL, 0000-0002-2245-0331

SARS-CoV-2 is a new pathogen responsible for the coronavirus disease 2019 (COVID-19) outbreak. Southeast Asia was the first region to be affected outside China, and although COVID-19 cases have been reported in all countries of Southeast Asia, both the policies and epidemic trajectories differ substantially, potentially due to marked differences in social distancing measures that have been implemented by governments in the region. This paper studies the across-country relationships between social distancing and each population's response to policy, the subsequent effects of these responses to the transmissibility and epidemic trajectories of SARS-CoV-2. The analysis couples COVID-19 case counts with real-time mobility data across Southeast Asia to estimate the effects of host population response to social distancing policy and the subsequent effects on the transmissibility and epidemic trajectories of SARS-CoV-2. A novel inference strategy for the time-varying reproduction number is developed to allow explicit inference of the effects of social distancing on the transmissibility of SARS-CoV-2 through a regression structure. This framework replicates the observed epidemic trajectories across most Southeast Asian countries, provides estimates of the effects of social distancing on the transmissibility of disease and can simulate epidemic histories conditional on changes in the degree of intervention scenarios and compliance within Southeast Asia.

## 1. Introduction

The politically and economically diverse region of Southeast Asia was the first region outside China to be affected by the coronavirus disease 2019 (COVID-19) pandemic, with early importations from Wuhan into Bangkok [1] and Singapore [2]. By mid-May 2020, over 50 000 cases of COVID-19, caused by the severe acute respiratory syndrome coronavirus 2 (SARS-CoV-2), have been reported across the ten countries that make up the Association of Southeast Asian Nations (Brunei, Cambodia, Indonesia, Laos, Malaysia, Myanmar, the Philippines, Singapore, Thailand and Viet Nam) [3]. Given the disparity in resources between these countries, as well as the age profile of their populations, there is potential for profound differences in the impact of the pandemic in this region, and the possibility that different approaches may be needed to mitigate its effects.

Infection with SARS-CoV-2 occurs primarily through respiratory droplets, facilitating rapid disease spread in close contact events at the household, workplace, school or wider community [4]; in Singapore, 71% of linked community COVID-19 cases were infected at home before the implementation of social distancing policies and 36% of such cases were infected at the workplace after

relaxation of social distancing policies [5]. Social distancing has thus emerged as a key element of most countries' early control options, an approach which seeks to reduce in number the instances of close contact in these settings, although policies of varying strictness were implemented across Southeast Asian countries [6]. Limited information exists, however, on how such reductions in social contacts influence transmissibility, which is commonly measured through the time-varying reproduction number $R_t$—defined as the ratio of new autochthonous cases and the total infection potential across all infected individuals at time $t$—in this region, which impedes decision making about what activities to curtail.

Although it shares 79.7% of its nucleotide identity with SARS-CoV, ongoing research suggests that SARS-CoV-2 may have a stronger interaction with the ACE2 reception binding domain, which leads to greater infectivity rates than has SARS-CoV [7]. Additionally, the structural coronavirus S-protein in SARS-CoV-2 contains a furin-like cleavage site, which may provide a gain-of-function and result in greater infectivity [8]. The transmission potential of SARS-CoV-2 is enhanced by a symptom profile which, for most patients, is much less severe than SARS-CoV, with fever, headache, dry cough and sore throat among the most common reported symptoms [9], as well as more specific symptoms such as anosmia [10,11].

In severe cases, however, approximately 20% of all hospitalized patients develop dysfunctional immune responses with massive proliferation of immune cells, overproduction of cytokines, and resulting myocardial damage and circulatory failure [12,13]. This response has been observed especially among the elderly and those with comorbidities [14–16], risk groups in which most deaths have occurred. This is a concern especially for Southeast Asia's ageing population [17], with COVID-19 mortalities in Southeast Asia exceeding 5000 in July 2020 [18]. Healthcare systems are put under strain as critically ill patients require long-term intensive care and medical supplies, and staff substantial protective equipment to ensure personal safety. Southeast Asian health systems have rather different critical care capacities, with over 10 beds per 100 000 population in Brunei, Singapore and Thailand but fewer than 3 in Indonesia, Laos, Myanmar and the Philippines [19].

To reduce epidemic growth and prevent rising death counts, governments worldwide, including in Southeast Asia, have implemented a mixture of social distancing measures, contact tracing and population-wide testing efforts [6]. In Viet Nam, whose early successes in controlling the epidemic are noteworthy [20], the early implementation of strict social distancing and lockdown of communities with outbreaks [21] has led to a plateau in reported incidence and subsequent easing of measures [20]. Manual contact tracing to identify and isolate cases has been used, for instance by Brunei [22] and Singapore [2]; both have repurposed other public servants such as the police to increase capacity. Meanwhile, temporary hospitals akin to China's Fangcang hospital model [23] have been established, for instance in exhibition centres in Malaysia [24] and Singapore [25], to isolate cases away from their families thus reducing the risk of household transmission [26]. Despite some initial successes using targeted approaches such as these, the countries of Southeast Asia have had to implement some form of generalized social distancing. For instance, in Malaysia a Movement Control Order was issued [24], but the number of cases did not fall enough for measures to be eased as quickly despite strict social distancing policies [27].

While mathematical modelling is mostly unanimous that social distancing may stem the short-term onward spread of SARS-CoV-2 [28–30], the observed efficacy of these measures is mixed, with varying country-specific epidemic trajectories before and after intervention implementation [3]. Policy-wise, governments have implemented a range of diverse movement suppression strategies, which primarily rely on public compliance for successful epidemic control [31]. Therefore, it is valuable to model the effects of implementation and compliance to these different intervention strategies between countries.

The time-varying reproduction number, $R_t$, has been used in dynamic transmission models of disease to quantify the evolving infection potential of viruses during epidemics, providing a metric to enable public health authorities to design, implement, and adapt interventions through the course of the epidemic [32]. Models of $R_t$ are highly relevant for the ongoing COVID-19 pandemic given the phased changes in control strategies across various locales [33,34] as well as spontaneous social distancing by the population. Through modelling $R_t$, [35] have shown that proper case identification together with isolation will stem the possible community spread of SARS-CoV-2. Additionally, [36] estimated the cluster-specific $R_t$ for SARS-CoV-2 in metapopulations and provided *post hoc* policy evaluation. These studies, however, estimate $R_t$ retrospectively and provide evaluations pre-post intervention strategies that have taken place. Heterogeneity in compliance to distancing measures, however, may substantively alter the subsequent effects of intervention measures on SARS-CoV-2 transmissibility and change the overall epidemic trajectories across countries. This epidemic marks the first pandemic in which social distancing can be objectively monitored through mobile telephony, though such data sources have been used to analyse or intervene in previous, smaller-scale epidemics [37,38]. To date, to the best of the authors' knowledge, no work has explicitly studied the across-country multiplicity in social distancing and each population's response to policy and the subsequent effects of these responses on transmissibility and the epidemic trajectories of COVID-19.

In this paper, we use publicly available reported COVID-19 case counts across nine Southeast Asian countries together with Google mobility data to infer the response of populations to social distancing policy and its subsequent effects on the transmissibility of SARS-CoV-2 and epidemic trajectories of COVID-19. We propose a novel inference strategy to relate the time-varying reproduction number explicitly to social distancing data through a regression structure. This framework is shown to replicate and fit the overall epidemic trajectories across most of Southeast Asia. It is also able to simulate alternative epidemic histories conditional on changes in the degree of intervention scenarios and/or compliance within each population. Using this new estimation methodology, we found significant between-country variations in the host population response to social distancing policy, variations in the effects of social distancing policy on the time-varying reproduction number and overall epidemic trends and lastly differences in the ability for even greater degrees of intervention within each country to stem the spread of SARS-CoV-2.

## 2. Methods

### (a) COVID-19 Data

Reported COVID-19 case counts from each Southeast Asian country were collected from the COVID-19 Data Github

Repository by the Center for Systems Science and Engineering at Johns Hopkins University where available. Case counts were extracted for 84 days from 15 February 2020 to 9 May 2020 for nine Southeast Asian countries with 655 million inhabitants, namely Cambodia, Indonesia, Laos, Malaysia, Myanmar, the Philippines, Singapore, Thailand and Viet Nam. Brunei was excluded for want of Google mobility data in the sultanate.

For the primary analysis, the serial interval distribution is taken from [39]. The incubation period distribution is taken from [40]. These data are used to estimate the infection potential and time-varying reproduction number of SARS-CoV-2 across countries. Sensitivity analysis on other SARS-CoV-2 serial interval and incubation period distributions was also conducted, with their values taken from literature. Full results are given in electronic supplementary material, technical appendix 1.

| parameter | value (days) | reference |
|---|---|---|
| serial interval | 4.6 (SD = 2) | [39] |
| incubation period | 5.6 (95% CI: 5.0, 6.3) | [40] |

## (b) Google mobility data

Mobility data were extracted from Google's COVID-19 Community Mobility Reports from 15 February 2020 to 9 May 2020 for the nine Southeast Asian countries aforementioned [41]. These data consist of percentage changes, compared to a baseline epoch before the pandemic, in time spent each day in six location types: residential, workplace, retail and recreation, grocery and pharmacy, parks, and transit stations, in each country. These data are generated from Android mobile telephone users who turn on their location history setting, and are aggregated and anonymized as described by the developers [41]. Mobile telephone penetration is high in most of Southeast Asia: above 100% for all countries except Laos (51%) and Myanmar (86%) [42]. Increasing time spent at home and reducing time spent at other locations is hypothesized to reduce the transmission potential of SARS-CoV-2 through a reduction in the number and diversity of social contacts [29,39]. To prevent collinearities obfuscating the signal, we focus on the change in time spent at home, using this feature in the model to estimate the effect of time-varying degrees of social distancing on the infection potential and time-varying reproduction number of SARS-CoV-2 across countries.

## (c) Policy information

Government and national news websites were sourced to collect the start times and types of social distancing measures implemented among the Southeast Asian countries studied. Social distancing measures have not been eased as of 20 May 2020 for all countries except Thailand and Viet Nam. The full source listing is provided in electronic supplementary material, technical appendix 2. These data are used to compare the magnitudes of policy intervention on observed social distancing.

## (d) Regression-augmented time-varying reproduction number

We define the time-varying reproduction number $R_t$ following [32] as the ratio of the number of new autochthonous cases reported on time $t$, $I_t$, and the total infection potential across all infected individuals at time $t$, $\Lambda_t$. In this formulation, $R_t$ can also be affected by other covariates $X_t$ observed over the same time window through a regression structure to be defined later, and is thus written $R_t(X_t)$ where needed for clarity. The infection potential across time is parameterized by the probability mass of the serial interval $w_s$ lasting $s$ time units,

$$\Lambda_t(w_s) = \sum_{s=1}^{t} I_{t-s} w_s. \tag{2.1}$$

Given the serial interval distribution $w_s$, data on the total number of incident cases $I_{0:t-1}$ and the reproduction number $R_t$ at time $t$, we have by definition the expected number of incident autochthonous cases,

$$\mathbf{E}(I_t | I_{0:t-1}, w_s, R_t, X_t) = R_t \Lambda_t. \tag{2.2}$$

A Poisson generative distribution for the number of local cases at time step $t$ is assumed, with the probability of observing $I_t^{\text{local}}$ cases at $t$ being

$$\mathbf{P}(I_t | I_{0:t-1}, w_s, R_t, X_t) \sim \text{Po}(R_t \Lambda_t(w_s)). \tag{2.3}$$

The baseline reproduction number is assumed to be constant [following 32] over the time period $[t - \tau, t]$ where $R_t$ is estimated. The probability of observing the local incidence $I_{[t-\tau,t]}$ given $R_t$ and lagged incidence data $I_{[0,t-\tau-1]}$ is given by

$$\mathbf{P}(I_{t-\tau t} | I_{0:t-\tau-1}, w_s, R_t, X_t) = \prod_{k=t-\tau}^{t} \frac{(R_t \Lambda_t(w_s))^{I_k} \exp(R_t \Lambda_t(w_s))}{I_k!}. \tag{2.4}$$

For convenience of exposition, we recast the scalars above in matrix form, namely $R_{[1,T]} = \mathbf{R}_{T \times 1}$ and other exogenous variables $X_{j,[1,T]} = \mathbf{X}_{j,T \times L}$. The following linear structure is placed on $\mathbf{R}$ and $\mathbf{X}_j$, the dependent and independent variables, respectively. We use $j$ to denote the types of exogenous variables considered. The independent variables in the regression equation consist of an intercept term and the percentage change, from the baseline epoch, in time spent in different locations—residential, workplace, retail and recreation, grocery and pharmacy, parks and transit stations—as taken from Google mobility data. We considered two paramerizations for $\mathbf{X}_j$, (i): with all Google mobility parameters and (ii): only the percentage change, from the baseline epoch, in time spent in residential address, due to potential multicollinearity among variables in (i). $\mathbf{X}_j$ is weighted by a kernel $\mathbf{K}_{1 \times L}$ parameterizing the effect of time decay of social distancing on $\mathbf{R}$. This is motivated by extant literature that infectiousness and time to symptom onset varies in time and among SARS-CoV-2 positive individuals [43]. $\mathbf{K}_{1 \times L}$ is a discretized version of the probability distribution of the incubation period. The discretization was conducted by taking 10 000 random draws from the estimated incubation period distribution in [40] and normalizing the sum of these draws over 1. Regression coefficients are denoted $\beta_j$, $j \in \{1 \cdots .P\}$ and independent Gaussian white noise with variance $\sigma^2$ by $\varepsilon_{T \times 1}$. $T$ denotes the number of maximum observations considered and $P$ the number of dependent variables in the regression equation. The specification can thus be given as

$$\mathbf{R} = \sum_{j=1}^{P} \mathbf{X}_j \mathbf{K} \beta_j + \epsilon = \mathbf{X}\beta + \epsilon. \tag{2.5}$$

The posterior distribution of $R_t$ can thus be defined given past incidence data $I_{[0,t]}$, the serial interval $w_s$, other exogenous variables $\mathbf{X}$, and regression parameters $\beta$, $\sigma$ and has no known conditionally conjugate form

$$\mathbf{P}(R_t | I_{0:t-\tau-1}, I_{t-\tau t}, w_s, \mathbf{X}, \beta, \mathbf{R}_{-t}, \sigma)$$
$$\propto \mathbf{P}(I_{t-\tau t} | I_{0:t-\tau-1}, w_s, \mathbf{R}, \mathbf{X}, \beta, \sigma) \mathbf{P}(\mathbf{R} | \mathbf{X}, \beta, \sigma)$$
$$\propto \left( \prod_{k=t-\tau}^{t} \frac{(R_t \Lambda_t(w_s))^{I_k} \exp(R_t \Lambda_t(w_s))}{I_k!} \right) \exp(-(\mathbf{R} - \mathbf{X}\beta)' \Sigma^{-1} (\mathbf{R} - \mathbf{X}\beta)). \tag{2.6}$$

## (e) Parameter estimation: sampling $\beta$ and $\sigma^2$

We perform Bayesian estimation for the parameters of interest following [32]. Priors are placed on parameters to obtain their respective conditionally conjugate distributions where possible, which were then sampled from, to obtain posterior draws. We estimate sequentially $\{\beta, \sigma^2\}$ by placing the following priors:

$$\beta \sim N(\beta_0, \Sigma_0),$$

with $\beta_0 = 0_{P \times 1}$ for $\beta$ to be centred around 0 and $\Sigma_0 = \text{diag}(100)_{P \times P}$ to impose a diffuse prior. $T_0 = \theta_0 = 1$ were used as hyperparameters for the prior of $\sigma$ to impose a diffuse prior distribution for $\sigma$

$$\sigma^2 \sim IG\left(\frac{T_0}{2}, \frac{\theta_0}{2}\right).$$

The conjugate posterior for $\beta$ can be sampled given $R$, $\mathbf{X}$, $\sigma$ from $\mathbf{N(\beta^*, \sigma^{*2})}$:

$$\beta^* = (\Sigma_0^{-1} + \frac{1}{\sigma^2}\mathbf{X'X})^{-1}(\Sigma_0^{-1}\beta_0 + \frac{1}{\sigma^2}\mathbf{X'R}), \tag{2.7}$$

$$\sigma^{*2} = (\Sigma_0^{-1} + \frac{1}{\sigma^2}\mathbf{X'X})^{-1}. \tag{2.8}$$

## (f) Parameter estimation: sampling $\mathbf{R}$

Unfortunately, Gibbs sampling between the Poisson distributed likelihood for $\mathbf{R} \in \{R_1 \ldots R_T\}$ and normally distributed prior is not possible

$$\mathbf{P}(R_t | I_{0:t-\tau+1}, I_{t-\tau:t}, w_s, \beta, \sigma, \mathbf{X})$$
$$\propto \mathbf{P}(I_{0:t-\tau+1} | R_t, I_{t-\tau:t}, w_s, \beta, \sigma, \mathbf{X})\mathbf{P}(R_t | \beta, \sigma, \mathbf{X}).$$

We used instead an autoregressive Metropolis–Hastings step with proposal distribution $R_t^{s+1} \sim N(R_t^s, V_R)$, where a draw $R_t^*$ from the proposal is accepted with probability

$$\min\left(1, \frac{\mathbf{P}(I_{0:t-\tau+1} | R_t^*, I_{t-\tau:t}, w_s, \beta, \sigma, \mathbf{X})\mathbf{P}(R_t^* | \beta, \sigma, \mathbf{X})}{\mathbf{P}(I_{0:t-\tau+1} | R_t^s, I_{t-\tau:t}, w_s, \beta, \sigma, \mathbf{X})\mathbf{P}(R_t^s | \beta, \sigma, \mathbf{X})}\right). \tag{2.9}$$

The Metropolis-within-Gibbs procedure [44] was conducted by iteratively sampling from the conditional posteriors (2.7)–(2.8) as well as the accept–reject step as described in (2.9). 10 000 draws were taken from each conditional posterior distribution with a burn-in of 5000 with Geweke convergence diagnostic checks [45] and visual inspection of trace plots conducted to ensure that the Markov chain Monte Carlo estimation described above is well behaved [44]. The fit of Google mobility data to the implied time-varying reproduction number in (2.5) and the time-varying reproductive structure on incidence data (2.2) were assessed across time visually and aggregated into goodness of fit summary statistics to ascertain model adequacy. Results of model fitting are provided in the electronic supplementary material, technical appendix 1.

## (g) Modelling intervention scenarios

We considered how changes in the levels of mobility from Google mobility categories over time led to a *post hoc* difference in the expected time-varying reproduction number and incidence across the time period where mobility data was captured. Briefly, we constructed a dampening scalar $\alpha_i \in \{0.1, 0.2 \ldots 0.9\}$ to the observed $\mathbf{X}$ and posterior mean estimate of state equation coefficients $\hat{\beta}$ to obtain the implied time-varying reproduction number $\hat{R}_i$ given the dampened level of mobility from that intervention scenario $i$

$$\hat{\mathbf{R}}_i = \alpha_i \otimes \mathbf{X}\hat{\beta}. \tag{2.10}$$

The expected reported incidence $\mathbf{E}(\hat{I}_{i,t}^{\text{local}} | I_{0:t-1,i}, w_s, \hat{R}_{i,t}, X_t)$ of COVID-19 under a specific scenario $i$ could then be regenerated given the contemporaneous infection potential $\Lambda_t$ and the implied scenario-specific time-varying reproduction number $\hat{R}_{i,t}$, through

$$\mathbf{E}(\hat{I}_{i,t}^{\text{local}} | I_{0:t-1,i}, w_s, \hat{R}_{i,t}, X_t) = \hat{R}_{i,t}\Lambda_t. \tag{2.11}$$

These scenarios' time-varying reproduction numbers and implied incidence were then compared against the baseline observed case wherein no dampening had taken place over the time period in which data were observed.

# 3. Results

## (a) Social distancing and the incidence of SARS-CoV-2

The incidence of diagnosed COVID-19 remained low throughout Southeast Asia before social distancing policies were implemented, with an average of 106.2 cases per day (figures 1 and 2).This ranged from 34.2 in Malaysia to 0 in Laos (figures 1 and 2). After implementation of social distancing, Southeast Asian countries saw an initial increase in average daily incidence of COVID-19 cases at an average of 1080 (figures 1 and 2). Singapore had the highest average daily case incidence of 520 (figures 1 and 2) while Laos continued to have a low case incidence at a daily average of 0.5 (figures 1 and 2). After two weeks of social distancing policy implementation, average daily incidence declined gradually for all countries except the Phillipines and Laos (figures 1 and 2).

Google mobility data demonstrated the changes in movement upon implementation of social distancing policies. The google mobility data included the following variables: percentage changes from baseline in movements in retail, grocery/pharmacy, parks, transit and residential locales. Before implementation of the social distancing policies in Southeast Asia, most countries there had little changes in baseline movement among the various localities, with changes from baseline mobility being close to zero. After implementation of the social distancing policies; however, all Southeast Asian countries saw a large decrease in their change from baseline in mobility across all localities outside the home. The greatest decrease from baseline mobility after policy implementation is in Malaysia at −47% while Viet Nam had the least decrease from baseline mobility after policy implementation, at −21%, though they arguably had less need for social distancing given the robustness of other aspects of their response (electronic supplementary material, technical appendix 1,2).

## (b) Social distancing and the time-varying reproduction number

Throughout Southeast Asia, there was a decrease in the average time-varying reproductive number before and after implementation of social distancing policies, from 2.08 (figure 3, 95% CrI: 1.80 to 2.38) to 1.33 (figure 3, 95% CrI: 1.17 to 1.49). There were, similarly, decreases in Indonesia, Thailand and Viet Nam after the implementation of social distancing policies. The greatest decrease was in Indonesia, where the mean time-varying reproductive number fell from 3.18 (figure 3, 95% CrI: 2.91 to 3.47) to 1.46 (figure 3, 95% CrI: 1.39 to 1.54) after implementation of social distancing—though it still lay above the epidemic threshold of unity. Cambodia, Laos and Myanmar are estimated, in contrast, to have had increases in the average $R_t$ before and after implementation of social distancing policies, the largest increase being seen in Myanmar from 0.59 (figure 3, 95% CrI:

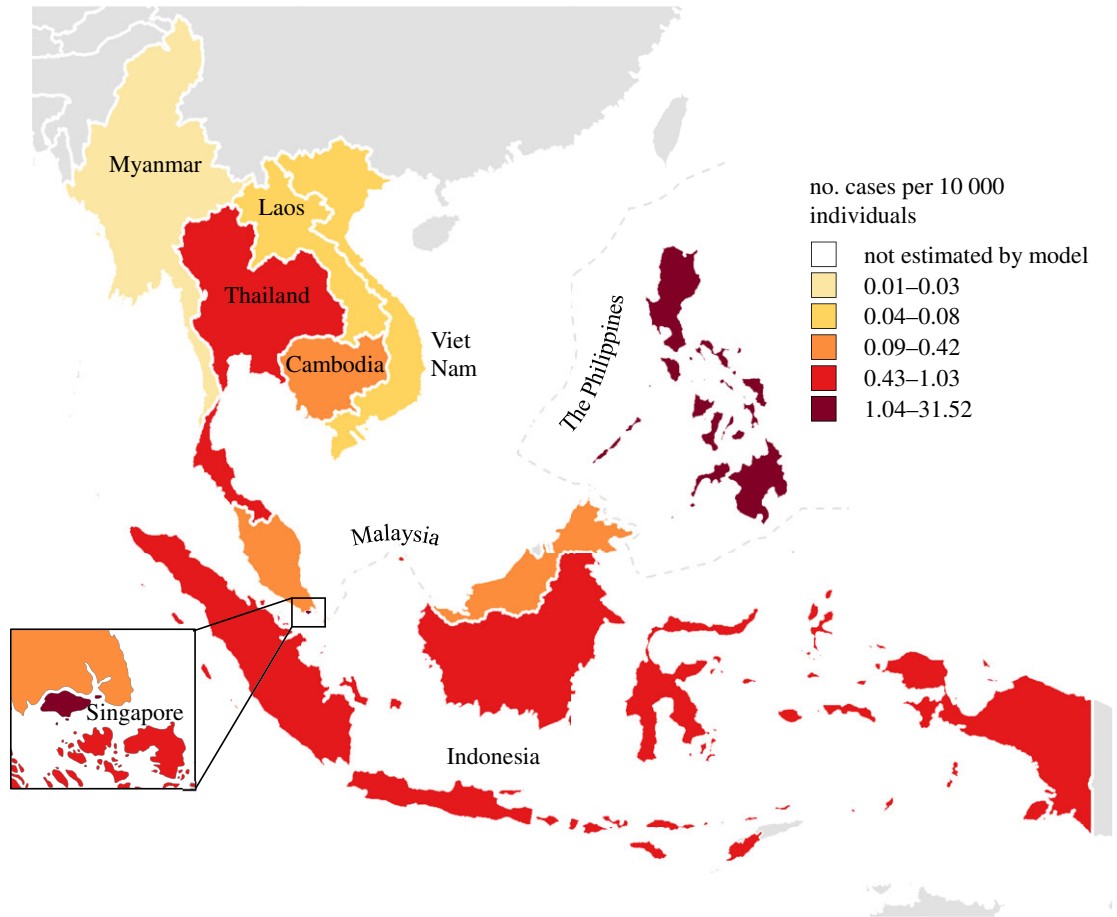

**Figure 1.** COVID-19 cumulative incidence per 10 000 across Southeast Asian countries from March 2020 to May 2020. (Online version in colour.)

0.22 to 0.95) to 1.07 (figure 3, 95% CrI: 0.85 to 1.30) after implementation of social distancing. Singapore, the Philippines and Malaysia saw relatively similar values of $R_t$ before and after implementation of social distancing policies.

## (c) Regional disparities in the effects of mobility on the transmission potential of SARS-CoV-2

In general, a positive change from baseline mobility in residences led to a negative effect on the time-varying reproduction number across countries. In all countries except Myanmar, coefficients of the changes in baseline mobility (CBM) from the residence category on the $R_t$ are negative, which means that a decrease from baseline levels of mobility from the residence category are associated to an expected decrease in $R_t$ (figure 4). Specifically, the largest effects of CBM in residences on the $R_t$ were evinced in Cambodia at −0.154 (95% CrI −0.1094 to −0.196), Thailand at −0.145 (95% CrI −0.128 to −0.161) and Indonesia at −0.109 (95% CrI −0.091 to −0.126): in all three countries, each 1% increase in time spent at home are associated with a reduction of at least 0.1 secondary case per index case. Malaysia (−0.073; 95% CrI −0.066 to −0.081), Viet Nam (−0.065; −0.046 to −0.084), the Philippines (−0.041; 95% CrI −0.030 to −0.051) and Laos (−0.037; 95% CrI −0.007 to 0.012) required more time at home to reach the same reduction in $R_t$. In Singapore, there was almost no effect on transmission overall (−0.009; 95% CrI −0.004 to −0.014), which may reflect that the model did not characterize the dynamics of $R_t$ well in the city state. Only Myanmar saw more transmissibility with more time spent in residences (0.033; 95% CrI 0.016 to

0.052). All coefficients had 95% credible intervals which exclude the null value except those for Laos.

## (d) Impact of increasing interventions on the transmission potential of SARS-CoV-2

We modelled how change in baseline mobility (CBM) from the residence category over time led to a *post hoc* difference in the expected time-varying reproduction number and incidence across the time period when mobility data were captured. Generally, simulating interventions by increasing the degree of time spent in residences, assuming the association with $R_t$ is maintained, leads to a decrease in the $R_t$ and concomitantly the incidence of SARS-CoV-2. Taking Indonesia as an example, a greater CBM for the residence category—from an extent of 100% to 190%—leads to a decrease in the mean $R_t$ from 5.63 to 2.93. Similarly, a larger CBM for the residence category from an extent of 100% to 190% leads to a decrease in average daily incidence from 309 to 997 (figure 3). In Singapore, a larger CBM for the residence category from an extent of 100% to 190% leads to a decrease in the mean $R_t$ from 2.65 to 1.95 and a decrease in incidence from 773.47 to 430.38. An exception to this trend would be Myanmar, where a larger CBM for the residence category from an extent of 110% to 190% leads to an increase in the mean $R_t$ from 0.48 to 1.00 and an increase in incidence from 0.17 to 2.50.

Excluding Myanmar, for which more time at home was associated with greater transmission, we investigated how much greater social distancing might be required to bring transmission below the epidemic threshold (i.e having the

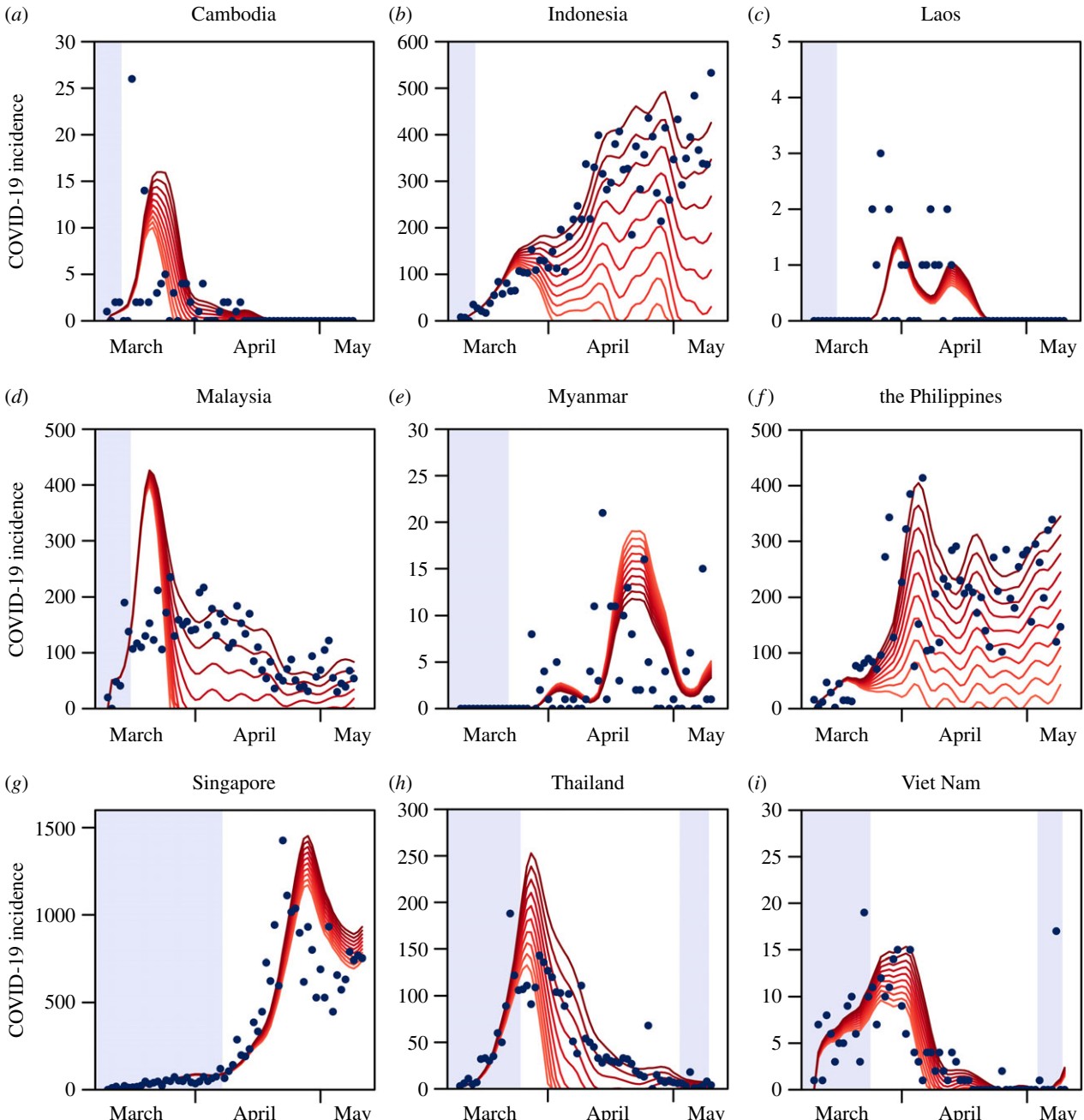

**Figure 2.** SARS-CoV-2 cases across time and time-varying reproductive number model fits across Southeast Asia from March 2020 to May 2020. Red lines represent the fitted case counts with lighter colours representing greater degrees of intervention. Blue points represent observed case counts. Highlighted areas represent time points where social distancing policies were not implemented. (Online version in colour.)

average $R_t$ to be below 1). A level of CBM for the residence category of 160–190% is required in Indonesia, Cambodia, Laos, Malaysia, the Phillipines, Thailand and Viet Nam.

## 4. Discussion

The COVID-19 pandemic poses enormous yet uneven challenges to the peoples of the world. In high-income countries, there has been a disproportional impact on the poor, on ethnic minorities, and on migrants [46–48]. Our results indicate large disparities across Southeast Asia in the time-varying reproduction number $R_t$ of SARS-CoV-2 between countries, with the largest average $R_t$ being 2.61 (for Malaysia) and the lowest, 0.83 (for Myanmar). Across Southeast Asia (except Brunei), over the 84 days in question, the average $R_t$ was 1.68, which is substantially lower than estimates of 2.35 for

the $R_t$ for Wuhan, China, where the pandemic had its genesis [35]. The post-intervention time-varying reproduction number remains, however, considerably higher than Kucharski et al's estimate of $R_t = 1.05$—i.e. to approximate epidemic equipoise—a week after Wuhan's unprecedented lockdown, and suggests that the efforts of most Southeast Asian countries are not sufficient to prevent exponential growth.

The large variations in social distancing policies across Southeast Asia led to marked differences in the reduction in reproduction numbers between countries, with the biggest decrease in Malaysia from 3.68 (95% CrI 3.47–3.91) to 1.53 (1.44–1.61) and the smallest decrease in Laos from 1.55 (1.04–2.08) to 1.20 (0.84–1.56). Our analysis finds these between-country differences in reductions in transmission risk can mostly be explained by the variability in time spent at home after lockdown policies were implemented. The countries of Southeast Asia implemented lockdown

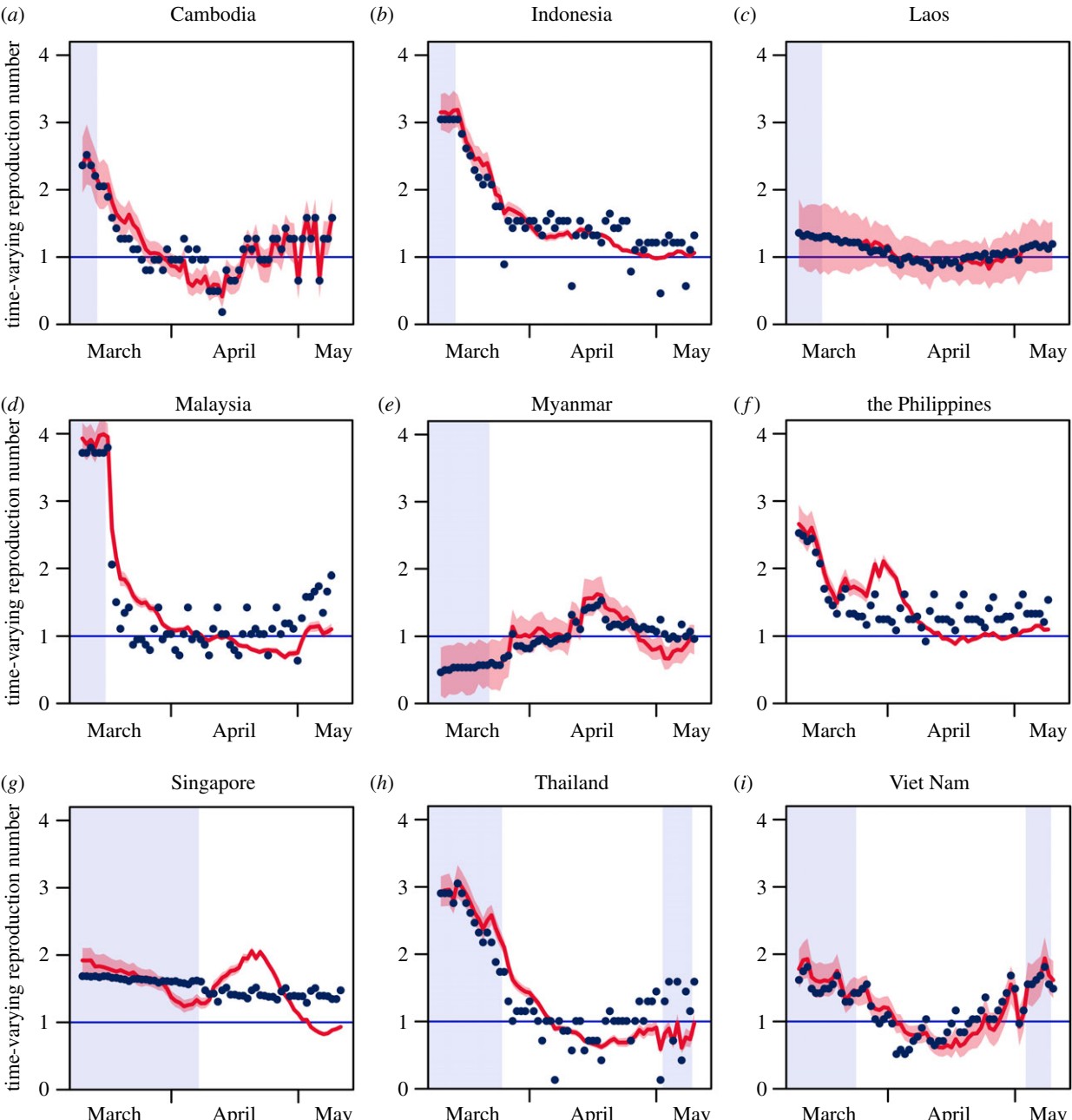

**Figure 3.** Estimated time-varying reproduction number for SARS-CoV-2 across time across Southeast Asian countries from March 2020 to May 2020. Red lines represent the fitted time-varying reproduction number with the shaded area representing the implied density estimate. Blue points represent observed time-varying reproduction number. Highlighted areas represent time points where social distancing policies were not implemented. (Online version in colour.)

differently, with for instance Malaysia implementing a relatively strict social distancing measures through a movement control order which was further enhanced to lockdowns in particular areas where incidence was high, while Laos—which has had few confirmed cases to date—only implemented residential lockdowns as the primary intervention for a shorter period of time (electronic supplementary material, technical appendix 2).

Three key reasons may explain this difference in effect sizes. First, the impact of policies to stay at home are crucially dependent on the number of essential workers who remain mobile during stricter social distancing periods [49]. Such exceptions to stay-at-home requirements may ameliorate the impact of social distancing policies by maintaining spread within essential-worker metapopulations and act as a bridge to household transmission [29]. In countries with larger families, which are the norm in many Asian countries, the reduction in $R_t$ caused by staying at home may be reduced

due to household sizes [50]. Finally, the degree of compliance and enforcement may vary greatly across the different Southeast Asian countries. The stringency of enforcement and population compliance towards periods of restricted movement may also dilute the treatment effects of social distancing policies on $R_t$ [51], where lower enforcement and compliance may further result in an increased risk for these individuals who do not comply despite stricter enforcement.

These results are in line with the extant body of work modelling the potential effects of social mixing on dampening the transmission potential of SARS-CoV-2 [15,29,30,35]. However, for Myanmar, we found that the effects of increasing time spent at home was associated with an increase the $R_t$. In general, due to ongoing transmission that is present in the community, reduction in non-home travel may not be as effective with the increased possibility of SARS-CoV-2 transmission being concentrated within residential localities. The implementation of stricter mobility laws across Southeast Asian

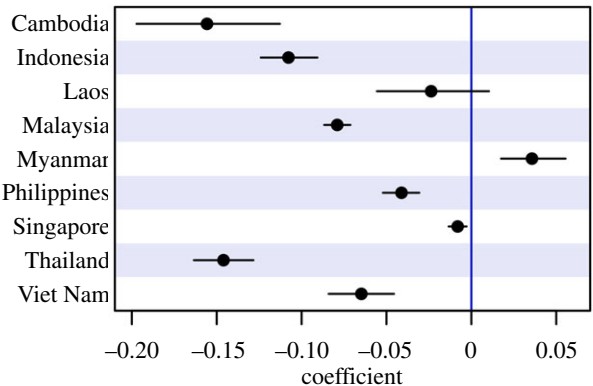

**Figure 4.** Estimated coefficient size and 95% credible intervals of changes from baseline in residences on the time-varying reproduction number across Southeast Asian countries. Points represent the posterior mean estimate, black lines represent the 95% credible intervals. Black lines which cross the threshold represented in blue signifies that the 95% credible interval contains 0. (Online version in colour.)

countries seeks to reduce widespread community spread [46]. However, for countries with already heightened community spread and presentation of unlinked cases, intervention measures may shift SARS-CoV-2 transmission patterns into residences [14].

There was a close correspondence between the modelled and estimated $R_t$ values (figure 3) for all countries except Singapore. Singapore's epidemic during the period of lockdown was concentrated in migrant workers living in large dormitory complexes, often living over ten to a room. In this environment, transmission may be facilitated by staying at 'home'. Repeating the Singapore analysis excluding cases in foreign worker dormitories (electronic supplementary material, technical appendix 1) reveals that transmission in the general community declined as time spent away from residential locations fell. The Singapore case shows that mobility data must be interpreted in tandem with traditional epidemiology.

There are several other limitations to our analysis. First, certain across-country variations are difficult to explicitly model in the $R_t$ framework. These include variations in reporting rate and potential epidemiological parameters which were taken from the literature (i.e. the serial interval). Sensitivity analysis in which other plausible serial interval distributions were used has, however, shown that there is little impact on the relationship between $R_t$ and covariates. Reporting rates may also vary across time, which may potentially trend our $R_t$ estimates in the same direction as the increase or decrease in reporting rates across time—this may be an issue as testing capacities

are exceeded. Importation incidence was also not accounted for in this study as such data were not consistently available. However, due to widespread travel bans in Southeast Asia early in the COVID-19 outbreak, only a small proportion of COVID-19 cases are imported, and these will not lead to sizeable errors in our estimates [32]. Lastly, although this study showed that increases in time spent in residences is associated with a decrease in reproduction number, this may not hold when countries lift social distancing measures [52].

Spatial variations within countries were not accounted for due to the paucity of geolocated COVID-19 data in Southeast Asia, but especially for larger and more heterogeneous countries in the region, such as Indonesia, such variability may be substantial. Future work should aim to estimate finer scale incidence data and plausibly examine the relationships between county specific interventions and their estimated $R_t$ and incidence.

Our study extends the little work conducted on the time-varying reproduction number of SARS-CoV-2 by providing explicit evaluation of this quantity before and after periods of social-distancing interventions across multiple countries. This study extends previously proposed methods to estimate the time-varying reproduction number [32,35,53] by allowing covariates to affect the $R_t$ and more importantly, statistical inference of covariate associations on the $R_t$, while accounting for uncertainty in parameters. Specifically, the methods developed in this paper can also account for time-varying movement patterns before and after policy implementation and thus provides valuable inference on the disparities in policy effects across Southeast Asian nations. The methods developed here are easily extendable to any infectious disease where covariates are posited to influence the time-varying reproduction number.

Data accessibility. All data used in the study are publicly available. Data and source are available as part of the electronic supplementary material

Authors' contributions. L.J.T., E.C.L.W., L.C.Z.X. and B.S.L.D. drafted the manuscript. L.J.T., B.S.L.D., J.K.R.H., M.P. and A.R.C. critically revised the manuscript. E.C.L.W. and L.C.Z.X collected the data. L.J.T. created the inference procedure and analysed the data. L.J.T., B.S.L.D. and A.R.C. visualized the results. L.J.T. conceived the study. A.R.C. provided supervision. All authors gave final approval for publication and agree to be held accountable for the work performed therein.

Competing interests. We declare we have no competing interest.

Funding. This work was supported by the Singapore Ministry of Health's National Medical Research Council under the Centre Grant Programme - Singapore Population Health Improvement Centre (NMRC/CG/C026/2017_NUHS) and grant no. COVID19RF-004.

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
