## [Reviewer comments · Proceedings of the Royal Society B: Biological Sciences]

Review History

RSPB-2020-1173.R0 (Original submission)

Review form: Reviewer 1

Recommendation

Accept with minor revision (please list in comments)

Scientific importance: Is the manuscript an original and important contribution to its field?

Excellent

General interest: Is the paper of sufficient general interest?

Excellent

Quality of the paper: Is the overall quality of the paper suitable?

Excellent

Is the length of the paper justified?

Yes

Should the paper be seen by a specialist statistical reviewer?

No

Do you have any concerns about statistical analyses in this paper? If so, please specify them explicitly in your report.

No

It is a condition of publication that authors make their supporting data, code and materials available - either as supplementary material or hosted in an external repository. Please rate, if applicable, the supporting data on the following criteria.

Is it accessible?

Yes

Is it clear?

Yes

Is it adequate?

Yes

Do you have any ethical concerns with this paper?

No

Comments to the Author

Please see the attached file for comments. (See Appendix A)

Review form: Reviewer 2

Recommendation

Major revision is needed (please make suggestions in comments)

Scientific importance: Is the manuscript an original and important contribution to its field?

Good

General interest: Is the paper of sufficient general interest?

Good

Quality of the paper: Is the overall quality of the paper suitable?

Good

Is the length of the paper justified?

Yes

Should the paper be seen by a specialist statistical reviewer?

No

Do you have any concerns about statistical analyses in this paper? If so, please specify them explicitly in your report.

No

It is a condition of publication that authors make their supporting data, code and materials available - either as supplementary material or hosted in an external repository. Please rate, if applicable, the supporting data on the following criteria.

Is it accessible?

Yes

Is it clear?

No

Is it adequate?

Yes

Do you have any ethical concerns with this paper?

No

Comments to the Author

The authors use a model that estimates the time-varying reproduction number to explore associations with mobility data in different Southeast Asian countries. This is an important question, and the modelling approach is appropriate. However, there are several points where the paper would benefit from some additional details and clarity.

I had the following comments:

- The authors quote the Ferguson et al study for the serial interval - is there a primary empirical data source they could refer to for this distribution? Similarly, the assumed incubation period is from Linton et al, and it would be worth confirming that this distribution is consistent with subsequently published estimates with large datasets.
- L139: It would be helpful to clarify the logic that time spent at home is being hypothesised to result in a reduction in social contacts (if, for example, most transmission was concentrated within the home, this would not necessarily be the case).
- L172: The "effect of time decay of social distancing" doesn't seem to be defined in the text. What is the motivation for this term?
- L218: The description of incidence rising after implementation of social distancing seemed a bit unclear, given the focus of the paper is on distancing and reduced transmission. It would be useful to have some more details in this paragraph about how to interpret this pattern.
- L282: "leads to" implies causality, but the model is exploring associations, so it would be more accurate to say something like "time spent in residences is associated with a decrease in the R_t "
- The Figure 2 caption needs more details, as it's not clear what the different lines represent. Likewise in Figure 3, it's not clear what the different points and lines are.
- L308: "the efforts of most Southeast Asian countries are not sufficient to prevent exponential growth" - this would be worth clarifying, as it's not clear this is the case in many countries currently?
- It would be worth mentioning in the discussion that correlation between mobility and transmission may not be the same as countries lift measures (e.g. <https://wellcomeopenresearch.org/articles/5-81>)

Minor comments:

- L278: SARS-CoV-2 needs to be capitalised.

Decision letter (RSPB-2020-1173.R0)

13-Jul-2020

Dear Mr LIM:

Your manuscript has now been peer reviewed and the reviews have been assessed by an Associate Editor. The reviewers' comments (not including confidential comments to the Editor) and the comments from the Associate Editor are included at the end of this email for your reference. As you will see, the reviewers and the Editors have raised some concerns with your manuscript and we would like to invite you to revise your manuscript to address them.

Research ethics:

Use of animals and field studies:

Please submit a copy of your revised paper within three weeks. If we do not hear from you within this time your manuscript will be rejected. If you are unable to meet this deadline please let us know as soon as possible, as we may be able to grant a short extension.

Best wishes,
Professor Hans Heesterbeek
mailto: proceedingsb@royalsociety.org

Associate Editor

Board Member: 1

Comments to Author:

The reviews are positive but with a large number of points for revision. I recommend that the authors carefully review all the recommended changes and clarifications before updating the manuscript accordingly.

Reviewer(s)' Comments to Author:

Referee: 1

Comments to the Author(s)

Please see the attached file for comments

Referee: 2

Comments to the Author(s)

The authors use a model that estimates the time-varying reproduction number to explore associations with mobility data in different Southeast Asian countries. This is an important question, and the modelling approach is appropriate. However, there are several points where the paper would benefit from some additional details and clarity.

I had the following comments:

- The authors quote the Ferguson et al study for the serial interval - is there a primary empirical data source they could refer to for this distribution? Similarly, the assumed incubation period is from Linton et al, and it would be worth confirming that this distribution is consistent with subsequently published estimates with large datasets.
- L139: It would be helpful to clarify the logic that time spent at home is being hypothesised to result in a reduction in social contacts (if, for example, most transmission was concentrated within the home, this would not necessarily be the case).
- L172: The "effect of time decay of social distancing" doesn't seem to be defined in the text. What is the motivation for this term?
- L218: The description of incidence rising after implementation of social distancing seemed a bit unclear, given the focus of the paper is on distancing and reduced transmission. It would be useful to have some more details in this paragraph about how to interpret this pattern.
- L282: "leads to" implies causality, but the model is exploring associations, so it would be more accurate to say something like "time spent in residences is associated with a decrease in the R_t ".
- The Figure 2 caption needs more details, as it's not clear what the different lines represent. Likewise in Figure 3, it's not clear what the different points and lines are.
- L308: "the efforts of most Southeast Asian countries are not sufficient to prevent exponential growth" - this would be worth clarifying, as it's not clear this is the case in many countries currently?
- It would be worth mentioning in the discussion that correlation between mobility and transmission may not be the same as countries lift measures (e.g. <https://wellcomeopenresearch.org/articles/5-81>)

Minor comments:

- L278: SARS-CoV-2 needs to be capitalised.

Author's Response to Decision Letter for (RSPB-2020-1173.R0)

See Appendix B.

Decision letter (RSPB-2020-1173.R1)

03-Aug-2020

Dear Mr LIM

I am pleased to inform you that your manuscript entitled "Revealing regional disparities in intervention on the transmission potential of SARS-CoV-2" has been accepted for publication in Proceedings B.

Open Access

Your article has been estimated as being 8 pages long. Our Production Office will be able to confirm the exact length at proof stage.

Paper charges

Sincerely,

Professor Hans Heesterbeek

Appendix A

This paper describes an analysis of the effects of SARS-CoV-2 interventions in South East Asian countries. The authors present an interesting and well written manuscript, which introduces a novel approach to infer the time-varying reproduction number and compares the results with the introduction of varying levels of distancing policy across SEA countries. I think this is a timely piece of work which I'm sure will be a strong contribution to our understanding of the COVID-19 pandemic and the effects of such distancing policies in practice.

I have several suggestions which I would recommend be addressed in the manuscript:

Line 29 – 117. Much of the introduction (paragraphs 2,3,4) focuses on SARS-CoV-2 overall, but it might be more relevant to focus more on the experience of COVID-19 in SEA specifically.

Line 127. It would be nice to have the serial interval and incubation period distributions written directly in the text for reference. The authors mention sensitivity analysis to these assumptions for the serial interval in the discussion – was this also done for the incubation period and are the results of this included in the appendices anywhere?

Line 182 - 213. I believe the first half of the methods section would be accessible to a wide audience, but from 'Sampling beta and σ^2 ' onwards, the methods perhaps become a bit less interpretable. The addition of some less technical interpretation could make this more accessible.

Line 219. The authors mention an increase in daily incidence after distancing was introduced. Would they imagine that this was mainly just due to the delay between a change in behaviour and this change being observed in case counts?

Line 226 - 236. The policy information is compared with the mobility data in text, but a figure with these overlaid might be interesting, to allow the reader to visualise the changes across countries.

Figures 2, 3, 4. I would prefer for the figure captions to be more informative and self-contained. e.g. state what the blue bars represent, what the blue points/red lines/error bounds represent. Perhaps a more informative label than 'coefficient' in fig 4.

Figure 3. Some of the incidence data (Malaysia, Philippines, Indonesia) appear to show ~weekly patterns in reporting – do the authors think that accounting for this could improve the fit?

Line 272 – 273. Does the Laos coefficient interval not include 0?

Line 297 – 378. I would find it interesting if the authors also included in the Discussion (or elsewhere if preferred), some discussion on the novelty of their inference approach, how this differs from other methods, and if they would foresee it being applicable more widely in the future.

Line 319. Could the authors comment more specifically on if those countries with stricter measures generally led to a greater reduction of R_t ? (Or if a mobility data figure is included, perhaps this will be interpretable by readers anyway).

A couple of typos noticed:

Line 26, provide(s). Line 155, data on (the) total. Line 338, missing a space 'timespent'.

Appendix B

Response to Reviewers on: *“Revealing regional disparities in the transmission potential of SARS-CoV-2 from interventions in South East Asia”*

Lim Jue Tao^{1,*}, Borame Sue Lee Dickens^{1,#}, Esther Choo Li Wen^{1,2,#},
Lawrence Chew Zheng Xiong^{1,3,#}, Joel Koo Rui Han¹, Clarence Tam¹,
Minah Park¹, Alex R Cook¹

¹Saw Swee Hock School of Public Health, National University of Singapore and National University Health System, Singapore

²Department of Biological Sciences, Faculty of Science, National University of Singapore

³Department of Geography, Faculty of Arts and Social Sciences, National University of Singapore

These authors contributed equally

* Corresponding author: Lim Jue Tao, ephljt@nus.edu.sg

July 17, 2020

Editor

Dear Mr LIM:

Your manuscript has now been peer reviewed and the reviews have been assessed by an Associate Editor. The reviewers' comments (not including confidential comments to the Editor) and the comments from the Associate Editor are included at the end of this email for your reference. As you will see, the reviewers and the Editors have raised some concerns with your manuscript and we would like to invite you to revise your manuscript to address them.

Research ethics:

Use of animals and field studies:

If your study uses animals please include details in the methods section of any approval and licences given to carry out the

study and include full details of how animal welfare standards were ensured. Field studies should be conducted in accordance with local legislation; please include details of the appropriate permission and licences that you obtained to carry out the field work.

If you wish to submit your data to Dryad (<http://datadryad.org/>) and have not already done so you can submit your data via this link, which will take you to your unique entry in the Dryad repository.

Electronic supplementary material: All supplementary materials accompanying an accepted article will be treated as in their final form. They will be published alongside the paper on the journal website and posted on the online figshare repository. Files on figshare will be made available approximately one week before the accompanying article so that the supplementary material can be attributed a unique DOI. Please try to submit all supplementary material as a single file.

Please submit a copy of your revised paper within three weeks. If we do not hear from you within this time your manuscript will be rejected. If you are unable to meet this deadline please let us know as soon as possible, as we may be able to grant a short extension.

Best wishes,
Professor Hans Heesterbeek

Response:

We thank the editors for the opportunity to revise our manuscript entitled “Revealing regional disparities in the transmission potential of SARS-CoV-2 from interventions in South East Asia” [RSPB-2020-1173]. We appreciate the constructive suggestions of both you and the reviewers.

Attached is a revised manuscript that we believe provides both clarity and cogency around the issues raised by the reviewers, and demonstrates the robustness of our approach. Below we summarize the salient issues raised by the reviewers, with our point-by-point responses. The corresponding changes in the manuscript are highlighted.

Thank you for your consideration of this revision.

Lim Jue Tao

Associate Editor

The reviews are positive but with a large number of points for revision. I recommend that the authors carefully review all the recommended changes and clarifications before updating the manuscript accordingly.

Response:

We thank the editor for the comment. All recommended changes below have been addressed and the manuscript has been updated accordingly.

Reviewer 1

Reviewer Point 1.1

This paper describes an analysis of the effects of SARS-CoV-2 interventions in South East Asian countries. The authors present an interesting and well written manuscript, which introduces a novel approach to infer the time-varying reproduction number and compares the results with the introduction of varying levels of distancing policy across SEA countries. I think this is a timely piece of work which I'm sure will be a strong contribution to our understanding of the COVID-19 pandemic and the effects of such distancing policies in practice. I have several suggestions which I would recommend be addressed in the manuscript:

Response:

We thank the reviewer for the kind comments. We have addressed the comments listed below and revised the manuscript accordingly with changes highlighted.

Reviewer Point 1.2

Line 29 – 117. Much of the introduction (paragraphs 2,3,4) focuses on SARS-CoV-2 overall, but it might be more relevant to focus more on the experience of COVID-19 in SEA specifically.

Response:

We thank the reviewer for the comment. We have incorporated and focused more on the experience of COVID-19 in SEA specifically, to frame the paper:

“...Infection with SARS-CoV-2 occurs primarily through respiratory droplets, facilitating rapid disease spread in close contact events at the household, workplace, school or wider community (Liu et al. 2020); in Singapore, 71% of linked community COVID-19 cases were infected at home before the implementation of social distancing policies and 36% of such cases were infected at the workplace after relaxation of social distancing policies (Khalik 2020). Social distancing has thus emerged as a key element of most countries' early control options, an approach which seeks to reduce in number the instances of close contact in these settings, although policies of varying strictness were implemented across Southeast Asian countries (Shang et al. 2020). Limited information exists, however, on how such reductions in social contacts influence transmissibility, which is commonly measured through the time varying reproduction number R_t —defined as the ratio of new autochthonous cases and the total infection potential across all infected individuals at time t —in this region, which impedes decision making about what activities to curtail....”

“...In severe cases, however, ~20% of all hospitalised patients develop dysfunctional immune responses with massive proliferation of immune cells, overproduction of cytokines, and resulting myocardial damage and circulatory failure (Mehta et al. 2020; Qin et al. 2020). This response has been observed especially among the elderly and those with comorbidities (Onder, Rezza, and Brusaferro 2020; Wu et al. 2020; Fang, Karakiulakis, and Roth 2020), risk groups in which most deaths have occurred. This is a concern especially for Southeast Asia's ageing population (World Bank 2016), with COVID-19 mortalities in Southeast Asia exceeding 5000 in July 2020 (Center for Strategic and International Studies 2020)....”

“...To reduce epidemic growth and prevent rising death counts, governments worldwide, including in Southeast Asia, have therefore implemented a mixture of social distancing measures, contact tracing, and population-wide testing efforts (Shang et al. 2020). In Viet Nam, whose early successes in controlling the epidemic are noteworthy (Pham et al. 2020), the early implementation of strict social distancing and lockdown of communities with outbreaks (Dreisbach 2020) has led to a plateau in reported incidence and subsequent easing of measures (Pham et al. 2020). Manual contact tracing to identify and isolate cases has been used, for instance by Brunei (Wong et al. 2020) and Singapore, (Pung et al. 2020); both have repurposed other

public servants such as the police to increase capacity. Meanwhile, temporary hospitals akin to China’s Fangcang hospital model (Chen et al. 2020) have been established, for instance in exhibition centres in Malaysia (Shah et al. 2020) and Singapore (Tee 2020), to isolate cases away from their families thus reducing the risk of household transmission (Dickens et al. 2020). Despite some initial successes using targeted approaches such as these, the countries of Southeast Asia have had to implement some form of generalised social distancing. For instance, in Malaysia a Movement Control Order was issued (Shah et al. 2020), but the number of cases did not fall enough for measures to be eased as quickly despite strict social distancing policies (Li 2020)....

Reviewer Point 1.3

Line 127. It would be nice to have the serial interval and incubation period distributions written directly in the text for reference. The authors mention sensitivity analysis to these assumptions for the serial interval in the discussion – was this also done for the incubation period and are the results of this included in the appendices anywhere?

Response:

We thank the reviewer for the comment. The serial interval and incubation periods used for the analysis are now attached in the main text:

“...For the primary analysis, the serial interval distribution is taken from Ferguson et al. 2020. The incubation period distribution is taken from Linton et al. 2020. These data are used to estimate the infection potential and time-varying reproduction number of SARS-CoV-2 across countries. Sensitivity analysis on other SARS-CoV-2 serial interval and incubation period distributions is also conducted, with their values taken from literature and full results given in the technical appendix...”

Parameter	Value (Days)	Reference
Serial Interval	4.6	Ferguson et al. 2020
Incubation Period	5.6 (95% CI: 5.0, 6.3)	Linton et al. 2020

Sensitivity analysis for the serial interval and incubation period used are also included within the appendix, with results included in Section 9,10. Sensitivity analysis indicates that our inference procedure is robust even with the serial interval or incubation period changed, with effect size estimates similar to the primary analysis in the main manuscript. The coefficient directions are also unchanged, with an increase in time spent in residences leading to a decrease in the time varying transmission potential of SARS-CoV-2 in most countries across all serial interval or incubation periods used in the sensitivity analysis. These results are presented in detail in the technical appendix and we urge the reviewer to refer to sections 9 and 10 for the changes.

Reviewer Point 1.4

Line 182 - 213. I believe the first half of the methods section would be accessible to a wide audience, but from ‘Sampling beta and sigma2’ onwards, the methods perhaps become a bit less interpretable. The addition of some less technical interpretation could make this more accessible.

Response:

We thank the reviewer for the comment. We have now first outlined our estimation procedure with a less technical interpretation:

“...We perform Bayesian estimation for the parameters of interest following Cori et al. 2013. Priors are placed on parameters to obtain their respective conditionally conjugate distributions where possible, which were then sampled from, to obtain posterior draws. We estimate sequentially $\{\beta, \sigma^2\}$ by placing the following priors...”

Reviewer Point 1.5

Line 219. The authors mention an increase in daily incidence after distancing was introduced. Would they imagine that this was mainly just due to the delay between a change in behaviour and this change being observed in case counts?

Response:

We thank the reviewer for the comment. It is possible that a delay in change in behaviour may have led to some increase in case counts after the implementation of social distancing policy. However, for most countries studied in this paper such as Singapore, Malaysia and Thailand, there is heavy enforcement at the start date of policy and the time varying reproduction is estimated to be lower compared to the periods before social distancing policy is implemented. The increase in cases after social distancing policy is thus more likely to be attributable to asymptomatic cases before social distancing which are symptomatic and reported after social distancing policy.

Reviewer Point 1.6

Line 226 - 236. The policy information is compared with the mobility data in text, but a figure with these overlaid might be interesting, to allow the reader to visualise the changes across countries.

Response:

We thank the reviewer for the comment. We have added an additional figure with the mobility data to guide the reader on mobility patterns in the technical appendix.

Reviewer Point 1.7

Figures 2, 3, 4. I would prefer for the figure captions to be more informative and self-contained. e.g. state what the blue bars represent, what the blue points/red lines/error bounds represent. Perhaps a more informative label than ‘coefficient’ in fig 4.

Response:

We have updated captions for figures 2–4 in the manuscript to be more informative and self-contained:

[Insert Figure 2 here]

Figure 2: SARS-CoV-2 cases across time and time-varying reproductive number model fits across Southeast Asia from March 2020 to May 2020. Red lines represent the fitted case counts with lighter colours representing greater degrees of intervention. Blue points represent observed case counts. Highlighted areas represent time points where social distancing policies were not implemented.

[Insert Figure 3 here]

Figure 3: Estimated time-varying reproduction number for SARS-CoV-2 across time across south-east Asian countries from March 2020 to May 2020. Red lines represent the fitted time-varying reproduction number with the shaded area representing the implied density estimate. Blue points represent observed time-varying reproduction number. Highlighted areas represent time points where social distancing policies were not implemented.

[Insert Figure 4 here]

Figure 4: Estimated coefficient size and 95% credible intervals of changes from baseline in residences on the time-varying reproduction number across south-east Asian countries. Points represent the posterior mean estimate, black lines represent the 95% credible intervals. Black lines which cross the threshold represented in blue signifies that the 95% credible interval contains 0.

Reviewer Point 1.8

Figure 3. Some of the incidence data (Malaysia, Philippines, Indonesia) appear to show weekly patterns in reporting – do the authors think that accounting for this could improve the fit?

Response:

We thank the reviewer for the interesting suggestion. The R_t inference procedure already accounts for contemporaneously evolving transmissibility of SARS-CoV-2 in the state equation, which then provides the expected number of cases on that date. Therefore, accounting for day of the week patterns was not necessary.

Reviewer Point 1.9

Line 272 – 273. Does the Laos coefficient interval not include 0?

Response:

We apologise for the typo and have rectified this expression accordingly:

“...All coefficients had 95% credible intervals which exclude the null value except those for Laos....”

Reviewer Point 1.10

Line 297 – 378. I would find it interesting if the authors also included in the Discussion (or elsewhere if preferred), some discussion on the novelty of their inference approach, how this differs from other methods, and if they would foresee it being applicable more widely in the future.

Response:

We thank the reviewer for the comment. An elaboration on the novelty of the inference approach, how it differs from other methods, and the applicability of the method to other diseases has been added to the discussion section:

“...Our study extends the little work conducted on the time-varying reproduction number of SARS-CoV-2 by providing explicit evaluation of this quantity before and after periods of social-distancing interventions across multiple countries. This study extends previously proposed methods to estimate the time-varying reproduction number (Cori et al. 2013; Kucharski et al. 2020; Thompson et al. 2019) by allowing covariates to affect the R_t and more importantly, statistical inference of covariate associations on the R_t , while accounting for uncertainty in parameters. Specifically, the methods developed in this paper can also account for time varying movement patterns before and after policy implementation and thus provides valuable inference on the disparities in policy effects across Southeast Asian nations. The methods developed here are easily extendable to any infectious disease where covariates are posited to influence the time-varying reproduction number....”

Reviewer Point 1.11

Line 319. Could the authors comment more specifically on if those countries with stricter measures generally led to a greater reduction of R_t ? (Or if a mobility data figure is included, perhaps this will be interpretable by readers anyway).

Response:

We thank the reviewer for the comment. Countries with generally stricter social distancing measures such as those observed in Malaysia and Thailand appear to correlate with a larger reduction in reproduction numbers compared to the rest of the region. An additional figure as suggested by Reviewer Point 1.8 now supplements the main manuscript, to guide the reader on how stricter measures led to a greater reduction of the R_t :

Change in baseline mobility in residences across 9 SEA countries. Time periods with no social distancing policy in place are indicated by gray regions.

Reviewer Point 1.12

A couple of typos noticed: Line 26, provide(s). Line 155, data on (the) total. Line 338, missing a space ‘timespent’.

Response:

We thank the reviewer for the comment. The typos have been rectified accordingly:

“...This framework replicates the observed epidemic trajectories across most Southeast Asian countries, **provides** estimates...”

“...Given the serial interval distribution w_s , data on **the** total number...”

“...Our analysis finds these between-country differences in reductions in transmission risk can mostly be explained by the variability in **time spent** at home...”

Reviewer 2

Reviewer Point 2.1

The authors use a model that estimates the time-varying reproduction number to explore associations with mobility data in different Southeast Asian countries. This is an important question, and the modelling approach is appropriate. However, there are several points where the paper would benefit from some additional details and clarity.

Response:

We thank the reviewer for the helpful comments. We hope that the revised manuscript addresses the recommendations provided by the reviewer and additional details and clarity surrounding our study.

Reviewer Point 2.2

The authors quote the Ferguson et al study for the serial interval - is there a primary empirical data source they could refer to for this distribution? Similarly, the assumed incubation period is from Linton et al, and it would be worth confirming that this distribution is consistent with subsequently published estimates with large datasets.

Response:

We thank the reviewer for this comment. We have conducted sensitivity analysis on the serial interval and also the incubation period using estimates from extant SARS-CoV-2 literature:

“...*For the primary analysis*, the serial interval distribution is taken from Ferguson et al. 2020. The incubation period distribution is taken from Linton et al. 2020. These data are used to estimate the infection potential and time-varying reproduction number of SARS-CoV-2 across countries. *Sensitivity analysis on other SARS-CoV-2 serial interval and incubation period distributions is also conducted, with their values taken from literature and full results given in the technical appendix...*”

Sensitivity analysis for the serial interval and incubation period used are also included within the appendix, with results included in Section 9,10. Sensitivity analysis indicates that our inference procedure is robust even with the serial interval or incubation period changed, with effect size estimates similar to the primary analysis in the main manuscript. The coefficient directions are also unchanged, with an increase in time spent in residences leading to a decrease in the time varying transmission potential of SARS-CoV-2 in most countries across all serial interval or incubation periods used in the sensitivity analysis. These results are presented in detail in the technical appendix and we urge the reviewer to refer to sections 9 and 10 for the changes.

The estimates used for the main manuscript also closely follow those in extant SARS-CoV-2 literature, as detailed in the technical appendix:

Serial Interval (days)	Reference
4.6 (SD=2)	Ferguson et al. 2020
3.96 (95% CI 3.53, 4.39)	Du et al. 2020
4.0 (95% CrI: 3.1, 4.9)	Nishiura, Linton, and Akhmetzhanov 2020
4.6 (95% CrI: 3.5, 5.9)	Nishiura, Linton, and Akhmetzhanov 2020
5.21 (95% CrI: -3.35, 13.94)	Ganyani et al. 2020
3.95 (95% CrI: -4.47, 12.51)	Ganyani et al. 2020

Serial interval estimates taken from literature

Incubation Period (days)	Reference
5.6 (95% CI: 5.0, 6.3)	Linton et al. 2020
5.1 (95% CI: 4.5, 5.8)	Lauer et al. 2020
6.4 (95% CrI: 5.6, 7.7)	Backer, Klinkenberg, and Wallinga 2020
4.9 (95% CI: 4.4, 5.5)	Jiang, Rayner, and Luo 2020
5.4 (95% CI: 4.8, 6.0)	Yang et al. 2020

Incubation period estimates taken from literature

Reviewer Point 2.3

L139: It would be helpful to clarify the logic that time spent at home is being hypothesised to result in a reduction in social contacts (if, for example, most transmission was concentrated within the home, this would not necessarily be the case).

Response:

We thank the reviewer for the comment. The logic that time spent at home is being hypothesised to result in a reduction in social contacts is clarified.

“..Increase in time spent at home is hypothesized to stave the transmission potential of SARS-CoV-2 through a reduction in social contacts (Prem et al. 2020; Ferguson et al. 2020). The change in time spent at home from these data is thus used in our model to estimate the effect of time-varying degrees of social distancing on the infection potential and time-varying reproduction number of SARS-CoV-2 across countries. ...”

Reviewer Point 2.4

L172: The ”effect of time decay of social distancing” doesn’t seem to be defined in the text. What is the motivation for this term?

Response:

We thank the reviewer for the comment. We have clarified the motivation of this term:

“... \mathbf{X}_j is weighted by a kernel $\mathbf{K}_{1 \times L}$ parameterising the effect of time decay of social distancing on \mathbf{R} . This is motivated by extant literature that infectiousness and time to symptom onset varies in time and among SARS-CoV-2 positive individuals (He et al. 2020)....”

Reviewer Point 2.5

L218: The description of incidence rising after implementation of social distancing seemed a bit unclear, given the focus of the paper is on distancing and reduced transmission. It would be useful to have some more details in this paragraph about how to interpret this pattern.

Response:

We thank the reviewer for the comment. We have elaborated and provided more details on the description of rising incidence and the subsequent decline in cases:

“...The incidence of diagnosed COVID-19 remained low throughout Southeast Asia before social distancing policies were implemented, with an average of 106.2 cases per day (Figure 1,2). This ranged from 34.2 in Malaysia to 0 in Laos (Figure 1,2). After implementation of social distancing, Southeast Asian countries saw an initial increase in average daily incidence of COVID-19 cases at an average of 1,080 (Figure 1,2). Singapore had the highest average daily case incidence of 520 (Figure 1,2) while Laos continued to have a low case incidence at a daily average of 0.5 (Figure 1,2). After two weeks of social distancing policy implementation, average daily incidence declined gradually for all countries except the Phillipines and Laos (Figure 1,2).”

Reviewer Point 2.6

L282: ”leads to” implies causality, but the model is exploring associations, so it would be more accurate to say something like ”time spent in residences is associated with a decrease in the R_t ”

Response:

We thank the reviewer for the comment. We have rectified “leads to” to “is associated with”.

“...Generally, simulating interventions by increasing the degree of time spent in residences is associated with a decrease in the R_t ...”

Reviewer Point 2.7

The Figure 2 caption needs more details, as it's not clear what the different lines represent. Likewise in Figure 3, it's not clear what the different points and lines are.

Response:

We have updated captions for figures 2–4 in the manuscript to be more informative and self-contained:

[Insert Figure 2 here]

Figure 2: SARS-CoV-2 cases across time and time-varying reproductive number model fits across Southeast Asia from March 2020 to May 2020. Red lines represent the fitted case counts with lighter colours representing greater degrees of intervention. Blue points represent observed case counts. Highlighted areas represent time points where social distancing policies were not implemented.

[Insert Figure 3 here]

Figure 3: Estimated time-varying reproduction number for SARS-CoV-2 across time across south-east Asian countries from March 2020 to May 2020. Red lines represent the fitted time-varying reproduction number with the shaded area representing the implied density estimate. Blue points represent observed time-varying reproduction number. Highlighted areas represent time points where social distancing policies were not implemented.

[Insert Figure 4 here]

Figure 4: Estimated coefficient size and 95% credible intervals of changes from baseline in residences on the time-varying reproduction number across south-east Asian countries. Points represent the posterior mean estimate, black lines represent the 95% credible intervals. Black lines which cross the threshold represented in blue signifies that the 95% credible interval contains 0.

Reviewer Point 2.8

L308: "the efforts of most Southeast Asian countries are not sufficient to prevent exponential growth" - this would be worth clarifying, as it's not clear this is the case in many countries currently?

Response:

We thank the reviewer for the comment. This expression refers to a cross country comparison pre-post social distancing policies in Wuhan, China versus Southeast Asia on estimates of the time varying reproduction number. We have reworded this sentence to clarify the point:

"...*The post policy time varying reproduction number* remains, however, considerably higher than Kucharski et al's estimate of $R_t = 1.05$ — i.e. to approximate epidemic equipoise— a week after Wuhan's unprecedented lockdown, and suggests that the efforts of most Southeast Asian countries are not sufficient to prevent exponential growth. ..."

Reviewer Point 2.9

It would be worth mentioning in the discussion that correlation between mobility and transmission may not be the same as countries lift measures (e.g. <https://wellcomeopenresearch.org/articles/5-81>)

Response:

We thank the reviewer for the comment. We have acknowledged that correlation between mobility and transmission may not be the same as countries lift measures:

"...*However, due to widespread travel bans in Southeast Asia early on in the COVID-19 outbreak, only a small proportion of COVID-19 cases are imported and will not lead to sizeable errors in our estimates (Cori et al. 2013). Lastly, although this study showed that increases in time spent in residences is associated with a decrease in reproduction number, this may not hold when countries lift social distancing measures (Ainslie et al. 2020)....*"

Reviewer Point 2.10

Minor comments: L278: SARS-CoV-2 needs to be capitalised.

Response:

We apologise for the typo and have rectified the capitalisation accordingly:

“...Impact of Increasing Interventions on the Transmission Potential of SARS-CoV-2”

References

- Ainslie, Kylie E C et al. (2020). “Evidence of initial success for China exiting COVID-19 social distancing policy after achieving containment”. In: *Wellcome Open Research* 5, p. 81. DOI: 10.12688/wellcomeopenres.15843.1.
- Backer, Jantien A, Don Klinkenberg, and Jacco Wallinga (2020). “Incubation period of 2019 novel coronavirus (2019-nCoV) infections among travellers from Wuhan, China, 20–28 January 2020”. In: *Eurosurveillance* 25.5. DOI: 10.2807/1560-7917.es.2020.25.5.2000062.
- Center for Strategic and International Studies (July 2020). *Southeast Asia Covid-19 Tracker*. URL: <https://www.csis.org/programs/southeast-asia-program/southeast-asia-covid-19-tracker-0>.
- Chen, Simiao et al. (2020). “Fangcang shelter hospitals: a novel concept for responding to public health emergencies”. In: *The Lancet* 395.10232, pp. 1305–1314. DOI: 10.1016/S0140-6736(20)30744-3.
- Cori, Anne et al. (2013). “A new framework and software to estimate time-varying reproduction numbers during epidemics”. In: *American Journal of Epidemiology*. DOI: 10.1093/aje/kwt133. URL: <https://doi.org/10.1093/aje/kwt133>.
- Dickens, Borame L et al. (2020). “Institutional, not home-based, isolation could contain the COVID-19 outbreak”. In: *The Lancet* 395.10236, pp. 1541–1542. DOI: 10.1016/S0140-6736(20)31016-3.
- Dreisbach, Jeconiah Louis (2020). “Vietnamese Public Health Practices in the Advent of the COVID-19 Pandemic: Lessons for Developing Countries”. In: *Asia Pacific Journal of Public Health*, p. 101053952092726. DOI: 10.1177/1010539520927266.
- Du, Zhanwei et al. (2020). “Serial Interval of COVID-19 among Publicly Reported Confirmed Cases”. In: *Emerging Infectious Diseases* 26.6, pp. 1341–1343. DOI: 10.3201/eid2606.200357.
- Fang, Lei, George Karakiulakis, and Michael Roth (2020). “Are patients with hypertension and diabetes mellitus at increased risk for COVID-19 infection?” In: *The Lancet Respiratory Medicine*. DOI: 10.1016/S2213-2600(20)30116-8.
- Ferguson, Neil et al. (2020). “Report 9: Impact of non-pharmaceutical interventions (NPIs) to reduce COVID19 mortality and healthcare demand”. In: *MRC Centre for Global Infectious Disease Analysis, COVID-19 Reports*. DOI: 10.25561/77482.
- Ganyani, Tapiwa et al. (2020). “Estimating the generation interval for coronavirus disease (COVID-19) based on symptom onset data, March 2020”. In: *Eurosurveillance* 25.17. DOI: 10.2807/1560-7917.es.2020.25.17.2000257.
- He, Xi et al. (2020). “Temporal dynamics in viral shedding and transmissibility of COVID-19”. In: *Nature Medicine* 26.5, pp. 672–675. DOI: 10.1038/s41591-020-0869-5.
- Jiang, Xuan, Simon Rayner, and Min-Hua Luo (2020). “Does SARS-CoV-2 has a longer incubation period than SARS and MERS?” In: *Journal of Medical Virology* 92.5, pp. 476–478. DOI: 10.1002/jmv.25708.
- Khalik, Salma (July 2020). *Workplace Covid-19 infections up after phase two reopening*. URL: <https://www.straitstimes.com/singapore/health/workplace-covid-19-infections-up-after-phase-two-reopening>.
- Kucharski, Adam J et al. (2020). “Early dynamics of transmission and control of COVID-19: a mathematical modelling study”. In: *The Lancet Infectious Diseases*. DOI: 10.1016/S1473-3099(20)30144-4.
- Lauer, Stephen A. et al. (2020). “The Incubation Period of Coronavirus Disease 2019 (COVID-19) From Publicly Reported Confirmed Cases: Estimation and Application”. In: *Annals of Internal Medicine* 172.9, pp. 577–582. DOI: 10.7326/m20-0504.
- Li, Toh Wen (May 2020). *Coronavirus: Singapore must tread carefully as circuit breaker measures are eased, says expert*. URL: <https://www.straitstimes.com/singapore/coronavirus-singapore-must-tread-carefully-as-circuit-breaker-measures-are-eased-says>.
- Linton, Natalie M et al. (2020). “Incubation period and other epidemiological characteristics of 2019 novel coronavirus infections with right truncation: a statistical analysis of publicly available case data”. In: *Journal of Clinical Medicine* 9.2, p. 538. DOI: 10.3390/jcm9020538.
- Liu, Ying et al. (2020). “The reproductive number of COVID-19 is higher compared to SARS coronavirus”. In: *Journal of Travel Medicine*. DOI: 10.1093/jtm/taaa021.
- Mehta, Puja et al. (2020). “COVID-19: consider cytokine storm syndromes and immunosuppression”. In: *The Lancet* 395.10229, pp. 1033–1034. DOI: 10.1016/S0140-6736(20)30628-0. URL: [https://doi.org/10.1016/S0140-6736\(20\)30628-0](https://doi.org/10.1016/S0140-6736(20)30628-0).
- Nishiura, Hiroshi, Natalie M. Linton, and Andrei R. Akhmetzhanov (2020). “Serial interval of novel coronavirus (COVID-19) infections”. In: *International Journal of Infectious Diseases* 93, pp. 284–286. DOI: 10.1016/j.ijid.2020.02.060.
- Onder, Graziano, Giovanni Rezza, and Silvio Brusaferro (2020). “Case-fatality rate and characteristics of patients dying in relation to COVID-19 in Italy”. In: *JAMA*. DOI: 10.1001/jama.2020.4683. URL: <https://doi.org/10.1001/jama.2020.4683>.
- Pham, Thai Quang et al. (2020). “The first 100 days of SARS-CoV-2 control in Vietnam”. In: *medRxiv*. DOI: 10.1101/2020.05.12.20099242.
- Prem, Kiesha et al. (2020). “The effect of control strategies to reduce social mixing on outcomes of the COVID-19 epidemic in Wuhan, China: a modelling study”. In: *The Lancet Public Health*. DOI: 10.1016/S2468-2667(20)30073-6.
- Pung, Rachael et al. (2020). “Investigation of three clusters of COVID-19 in Singapore: implications for surveillance and response measures”. In: *The Lancet*. DOI: 10.1016/S0140-6736(20)30528-6.
- Qin, Chuan et al. (2020). “Dysregulation of immune response in patients with COVID-19 in Wuhan, China”. In: *Clinical Infectious Diseases*. DOI: 10.1093/cid/ciaa248.

- Shah, Ain Umaira Md et al. (2020). “COVID-19 outbreak in Malaysia: Actions taken by the Malaysian government”. In: *International Journal of Infectious Diseases* 97, pp. 108–116. DOI: 10.1016/j.ijid.2020.05.093.
- Shang, Jian et al. (2020). “Structural basis of receptor recognition by SARS-CoV-2”. In: *Nature*, pp. 1–4. DOI: 10.1038/s41586-020-2179-y. URL: <https://doi.org/10.1038/s41586-020-2179-y>.
- Tee, Zhuo (Apr. 2020). *S’pore Expo 2nd facility for community isolation*. URL: <https://www.straitstimes.com/singapore/health/spore-expo-2nd-facility-for-community-isolation>.
- Thompson, RN et al. (2019). “Improved inference of time-varying reproduction numbers during infectious disease outbreaks”. In: *Epidemics* 29, p. 100356.
- Wong, Justin et al. (2020). “Responding to COVID-19 in Brunei Darussalam: Lessons for small countries”. In: *Journal of Global Health* 10.1. DOI: 10.7189/jogh.10.010363.
- World Bank (2016). *Live Long and Prosper: Aging in East Asia and Pacific*. URL: <http://pubdocs.worldbank.org/en/165351470911396346/Live-Long-and-Prosper.pdf>.
- Wu, Joseph T et al. (2020). “Estimating clinical severity of COVID-19 from the transmission dynamics in Wuhan, China”. In: *Nature Medicine* 26.4, pp. 506–510. DOI: 10.1038/s41591-020-0822-7.
- Yang, Lin et al. (2020). “Estimation of incubation period and serial interval of COVID-19: analysis of 178 cases and 131 transmission chains in Hubei province, China”. In: *Epidemiology and Infection* 148. DOI: 10.1017/s0950268820001338.